# Plastic Hotspot Mapping in Urban Water Systems

**Paolo Tasseron [1,†], Hestia Zinsmeister [1,†], Liselotte Rambonnet [2,3] , Auke-Florian Hiemstra [3,4], Daniël Siepman [3] and Tim van Emmerik [1,*]**

[1]  Hydrology and Quantitative Water Management Group, Wageningen University and Research, 6708 PB Wageningen, The Netherlands; paolo.tasseron@wur.nl (P.T.); hestia.zinsmeister@wur.nl (H.Z.)
[2]  Science Communication and Society, Leiden University, Rapenburg, 70, 2311 EZ Leiden, The Netherlands; l.rambonnet@biology.leidenuniv.nl
[3]  De Grachtwacht, Lange Mare 25, 2312 GP Leiden, The Netherlands; aukeflorian.hiemstra@naturalis.nl (A.-F.H.); mail@danielsiepman.nl (D.S.)
[4]  Naturalis Biodiversity Center, Darwinweg 2, 2333 CR Leiden, The Netherlands
*  Correspondence: tim.vanemmerik@wur.nl
†  These authors contributed equally to this work.

**Abstract:** Reducing plastic pollution in rivers, lakes, and oceans is beneficial to aquatic animals and human livelihood. To achieve this, reliable observations of the abundance, spatiotemporal variation, and composition of plastics in aquatic ecosystems are crucial. Current efforts mainly focus on collecting data on the open ocean, on beaches and coastlines, and in river systems. Urban areas are the main source of plastic leakage into the natural environment, yet data on plastic pollution in urban water systems are scarce. In this paper, we present a simple method for plastic hotspot mapping in urban water systems. Through visual observations, macroplastic abundance and polymer categories are determined. Due to its simplicity, this method is suitable for citizen science data collection. A first application in the Dutch cities of Leiden and Wageningen showed similar mean plastic densities (111–133 items/km canal) and composition (75–80% soft plastics), but different spatial distributions. These observations emphasize the importance of long-term data collection to further understand and quantify spatiotemporal variations of plastics in urban water systems. In turn, this will support improved estimates of the contribution of urban areas to the plastic pollution of rivers and oceans.

**Keywords:** macroplastic; citizen science; plastic soup; The Netherlands; hydrology; observations; plastic pollution; marine litter

## 1. Introduction

Plastic pollution in aquatic environments negatively impacts human livelihood and ecosystem health. Around 69% of all plastic waste ever generated has ended up in landfills or the natural environment [1]. Land-based plastic waste is assumed to be an important source of marine plastic pollution. Rivers play a key role in the transport of plastics from source to sea and are estimated to emit around 1 Mt of plastics into the ocean annually [2]. Yet, even in some of the world's most polluted river systems, the total plastic emission to other water bodies is only 3% of the plastic waste generated within the river basin [3]. The remaining plastic waste accumulates within the river system—on riverbanks, in the sediment, or clogging hydraulic infrastructure [4].

Urban areas play an important role in plastic pollution, because densely populated areas, especially in regions with lacking waste infrastructure, are associated with high leakage rates of plastics into the environment [5]. This makes urban areas one of the main sources of plastic pollution in aquatic ecosystems. However, plastic waste also directly affects urban water systems negatively. Plastic waste clogs urban drainage infrastructure and causes a larger and faster increase in water level compared to

blockage by organic materials [6]. These blockages increase the flood extent and depth around the blockage locations [7]. Plastic litter also threatens aquatic life through ingestion and entanglement. In French and Swiss freshwater bodies, up to 12% of the analyzed birds and fish were found to contain plastics [8]. Despite urban areas being main sources of plastic pollution and the clear direct negative effects, data on the abundance of plastics in urban water systems are scarce.

In recent years, there has been an increased effort to collect data on the abundance of macroplastics in river systems. Sampling locations further away from urban areas are often found to have less or smaller plastic litter items [9,10]. For the Seine (France), Gasperi et al. [11] analyzed waste collected from 26 floating booms, confirming the urban areas as the main source of river plastic pollution. Schirinzi et al. [12] found that for the Llobregat and Besòs rivers flowing through Barcelona (Spain), precipitation events caused an additional influx of plastic litter from urban riverbanks. Lahens et al. [13] sampled plastic in urban canals in Ho Chi Minh City (Vietnam) and demonstrated that urban canals have a much higher plastic fragment concentration than the Saigon river at the outlet of the canal. Recent plastic data for rivers flowing through Jakarta (Indonesia) and Manila (Philippines) revealed a clear spatial variation of plastic within urban water systems. These studies show the value of observational data, as they could be used to detect point sources (drains, sewages, waste dumps) of riverine plastic waste [3,14]. Other initiatives include citizen science or voluntary cleanup actions, such as canoeing sessions for waste collection in the city of Leiden, the Netherlands. These efforts can give good qualitative information on the litter types, including plastic items abundant in urban water systems (see Figure 1 for litter items found in Leiden), but often do not yield quantitative data. For a better understanding of the abundance, sources, sinks, and transport mechanisms of plastic in urban water systems, a harmonized method for data should be developed.

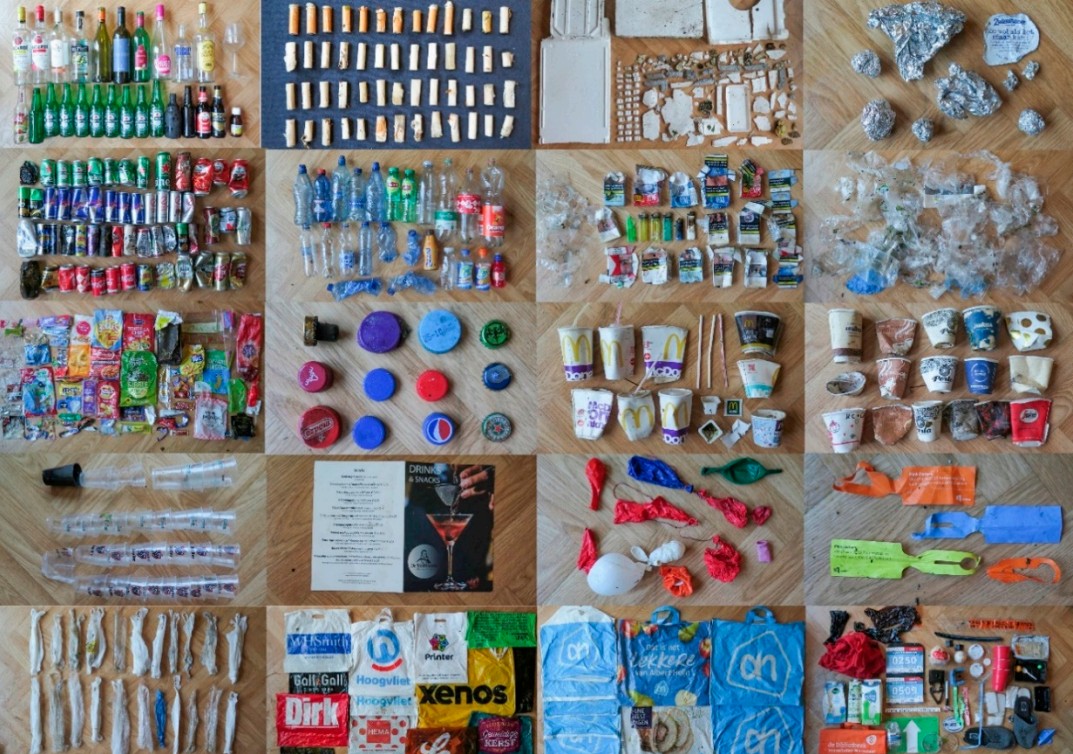

**Figure 1.** All litter encountered during a two-hour canoeing session at a Plastic Spotter cleanup in the canals of Leiden, the Netherlands, 21 December 2019 (credit: Daniël Siepman).

This paper presents a simple method for hotspot mapping of plastic waste in urban water systems. The method includes observations of floating plastic item transport, density, and polymer category. By defining and prioritizing strategic locations, a first-order estimate of the spatial distribution of

plastic waste in urban water systems can be made. These data can be used by policymakers and other stakeholders to identify plastic waste sources, optimize urban waste infrastructure, and improve cleanup efforts. The latter includes identifying the most effective cleanup routes for mobile cleanup initiatives (e.g., Plastic Spotter Canoe Fleet; Citizen Science Lab, [15]) or permanent waste collection devices (e.g., Seabin Project [16]) or The Great Bubble Barrier [17]). As no additional equipment is required, the method can also be used for data collection through citizen science, such as done previously for German and Dutch riverbanks [18,19]. Data collection can be done using the mobile citizen science app CrowdWater, which includes a module for measuring plastics in urban and natural water systems [20]. The method also allows for frequent repetition, to collect data over longer periods. Time-varying hotspot maps can, in turn, be used for investigating natural effects including wind, water current, and seasonal variation, and anthropogenic effects such as specific events and policy changes.

## 2. Methods

We propose a simple method to identify floating macroplastic hotspot locations in urban water systems. The proposed strategy can easily be combined with various plastic monitoring and sampling methods. Existing contemporary initiatives mainly focus on cleanups by kayaks or canoes [21], floating debris-retention booms [11], citizen science data collection [18], litter surveys along riversides, spectral data analyses on bridges, and unmanned aerial vehicles (UAVs) [22]. Through the strategy presented in this paper, data obtained through different methods can be used to quantify macroplastic distribution in urban water systems and identify hotspots. In the following, we first discuss the used data collection technique. After that, we introduce the seven-step plastic hotspot method and discuss its application in the cities of Leiden and Wageningen.

### 2.1. Visual Observations

To determine the location and magnitude of plastic hotspots, data were collected by visual counting of floating plastic in the study areas. This method is based on the method proposed by González-Fernández and Hanke [23] for quantifying floating litter in rivers. As the flow in urban water systems is often minimal, we adopted this method. The urban water system was divided into 5-m long stretches, and all visible floating and superficially submerged plastics (estimated up to 10 cm depth) were counted across the entire width (Supplementary Table S1). Plastics on the banks or quays were not measured in this research. The counted plastic items were also categorized in the seven different plastic categories (Figure 2), which will be discussed in more detail in the next section. Locations where no plastic debris was observed were marked as clean, appearing in the dataset as a zero measurement. In the next paragraph, the seven-step methodology is presented, including practical trade-offs that should be considered for its application.

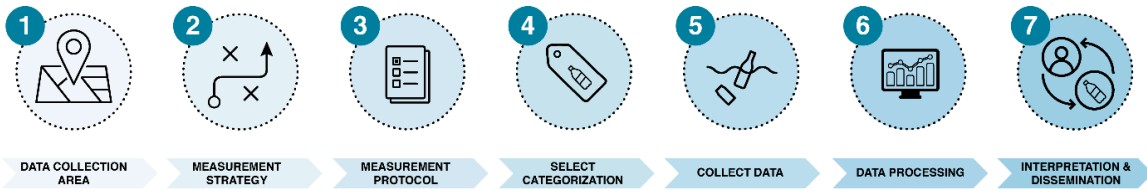

**Figure 2.** The seven steps for plastic hotspot mapping in urban water systems.

### 2.2. Seven Steps for Plastic Hotspot Mapping

The plastic hotspot mapping method consists of seven steps, as illustrated in Figure 2. This method aims to give guidance for harmonized measurements in urban water systems, but individual choices for each step can be tuned to cater towards local research questions or varying circumstances. Each step is presented and discussed below.

### 2.2.1. Defining the Data Collection Area

The first point of decision in the data collection process is the identification of the research area. Urban water systems can be rather complex, and it is important to clearly scope the area of interest. In the cases of Leiden and Wageningen, the size of the city and extent of the canals determined the area that could be covered in one day. Once multiple-day observations are required for constructing single plastic hotspot maps, this may lead to plastics being counted twice or other inaccuracies. For long-term monitoring, this is no issue, as hotspot maps can be compared for different moments in time to explore spatiotemporal variations. Obtaining information and expertise of local initiatives and experts can be crucial in this decision process. Our method aims to provide data for the most important stretches of an urban water system within a working day. Therefore, the focus of data collation was only on plastic debris in the canals, and plastic on canal quays was excluded.

### 2.2.2. Measurement Strategy

For data collection of plastic debris in an urban water system, it should be determined whether to observe the canals from one or two sides. Two-sided surveys require more time, but have the additional benefits of (1) allowing the comparison of the abundance at each side of the canal, and (2) increasing the likelihood of items being observed if the canal is relatively wide or when the view is blocked by (house) boats. For this study, counting plastic items by walking transects along one side of the canal was selected as the measurement strategy. One side of each selected canal was chosen, aiming to collect data over the whole width of the canal by visually counting the items. This was only possible when the view over the canal was not obstructed, for instance, the presence of recreational boats and fixed houseboats. This strategy was possible for Leiden and Wageningen because the canal widths did not exceed 40 m and 20 m, respectively. For measuring urban water systems with wider waterways (e.g., Amsterdam, Jakarta, Manila), two-sided surveys are more accurate. Alternatively, observations can be done from bridges, during which the canal width can be divided into segments of 5 to 15 m (van Emmerik et al., 2019).

### 2.2.3. Measurement Protocol

Preliminary to the data collection in the field, a measurement protocol was defined. This protocol defined the conditions of each measurement to ensure a coherent dataset. All canal segments were marked as a measurement point, wherein an area of 5 m along the canal length and over the whole canal width, items were counted. If no plastics were counted within a canal segment, the area is clear and therefore, marked as a measurement point with a zero plastic count. In several cases when plastic items were mobile during a measurement, the item was counted in the segment that was observed at that moment. This decision was made because of the slow water current in both cities, and therefore, it was too difficult to properly include floating rate and other variables in the research. Lastly, if the source of the plastic debris was evident, we recorded this in the field notes (e.g., an open container at a construction site next to the canal). From every measurement location, the debris was documented by counting the items and making photographs.

### 2.2.4. Categorization

At each location, the plastic items were categorized into seven different categories, respectively: polyethylene terephthalate (PET), soft polyolefins (PO-soft), hard polyolefins (PO-hard), multilayer, polystyrene (PS), expanded polystyrene (EPS), and other, see Figure 3. These categories are based on the structural characteristics of the different types of plastic. Per location, the total count per category was recorded. Collecting data about the different types of plastic found is important to track the source and to understand the consequences it might have. For instance, different traits influence how long it will take for plastic to fragment into smaller pieces in water systems (van Emmerik and Schwarz, 2020). Note that the actual categorization protocol can be changed according to the research question or goal

of the plastic hotspot mapping. Examples of other categorization protocols can be found in Vriend et al. (2020).

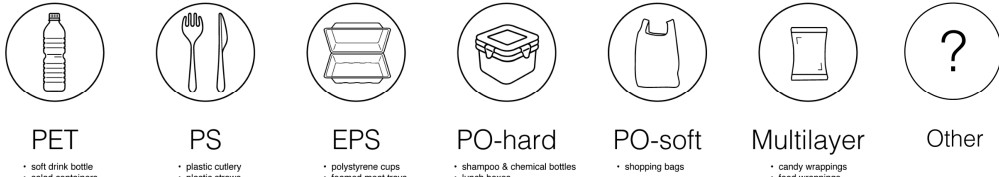

**Figure 3.** The categories used for classification in this research, including some typical items that belong to these classes. Items that cannot be identified as one of the six specific classes are categorized as "other".

### 2.2.5. Data Collection

The actual data collection can be done ad hoc, or through existing data collection tools. In this study, we did not use existing data collection tools, as we focused on developing a general plastic hotspot mapping method. The method has been developed in such a way to facilitate easy integration with the CrowdWater mobile app for data collection [20]. CrowdWater can be used for measurements of amount and categories of plastic items in urban and natural water systems. Measurement can be done by either creating a new location, or by adding a measurement to an existing location. This allows for both building long-term datasets at specific locations, such as hotspots, or introducing new locations in the overall data collection strategy.

### 2.2.6. Data Processing

Several steps were taken to create the plastic hotspot maps, which are shortly summarized below. The exact steps taken to process the data will differ depending on the choices in the previous five steps, but will follow the same general procedures as described next. Each data point was recorded with a latitude and longitude, date, and time, providing the essentials for creating the plastic hotspot maps. First, the sum of items of each plastic category was taken, and its percentage of the total number of items. Secondly, the hotspot map was created based on the coordinates and the associated count of all plastic items per 5-m segment. These were subsequently categorized into five different ranges. Values higher than 4 items/m were considered hotspots, as no common baseline values are available. For comparison with item density in rivers and other aquatic ecosystems, the item density per kilometer was determined by dividing the total count for the whole system by the length of the canals in the associated study area.

### 2.2.7. Interpretation and Dissemination

Lastly, it is essential to interpret the movement of plastics in surface water from the scope of the local urban system properties to make a connection with behavior of citizens and pollution patterns. Different categories of plastic are associated with variations in size and density which influence flow dynamics. If the type of plastic and source of plastic in the system is known, a more accurate description and interpretation of the data is feasible. Consistent dissemination of the results and key findings to stakeholders, such as the municipality, the data collection volunteers, and citizens is crucial to optimize plastic leakage interventions, data collection protocols, cleaning efforts, and increase awareness [24]. This can be done through clear hotspot maps, such as presented in this paper. These maps can be made available through project websites (e.g., CrowdWater [20], Plastic Spotter [15]), but may also be presented in an online dashboard hosted by the municipality.

### *2.3. Case Studies in Leiden and Wageningen, The Netherlands*

Two Dutch cities were selected as study areas for this research—Leiden and Wageningen. Leiden was selected because of the ongoing Plastic Spotter project [15], which aims to monitor and reduce plastic pollution in the urban water system. Wageningen was selected for practical reasons. Leiden (52°9′ N, 4°29′ E) is in the province of South Holland and has 125,000 inhabitants. The urban water system in Leiden is 28 km long and consists of an outer canal surrounding the historic city center. East of Leiden, the Old Rhine River splits in two. Both branches enter the city, where they merge and transverse the urban water system. Wageningen (51°58′ N, 5°40′ E) is in the province of Gelderland and has 39,000 inhabitants. The current urban water system is 1.8 km long and consists of an outer canal surrounding the historical city center, and is drained to the Nederrijn river. The entire 1.8 km of canals in Wageningen was selected as the data collection area. For Leiden, data were collected for 3.9 km of the urban water system, with a main focus on the canals in the historic city center. The sampling was conducted on two different days: Thursday 5 March 2020 for Wageningen and Friday 6 March 2020 for Leiden. Both days had similar weather conditions and no precipitation was recorded.

## 3. Results and Discussion

### *3.1. Plastic Hotspot Maps in Leiden and Wageningen*

The survey revealed four clear hotspots (> 4 items/m) in the urban water system of Leiden (Figure 4). The hotspots were found in the north, east, west, and south of the city. Plastic densities between 0 and 4 items/m were found across the whole urban water system. Increased densities (2–4 items/m) were found in the middle of the city, close to the open market. In Wageningen, fewer plastic hotspots were found, and only on the northwest side of the urban water system (Figure 5). In the other stretches of the system, only three points with plastic densities larger than zero were found. The hotspots are observed to be related to the parking areas and recreational areas along this stretch of the canal system, which are not present to the same extent in the other sections of the canal system. Wind speed and direction may also influence the distribution of hotspots, but more observations are needed to further investigate this. Despite the differences in the spatial distribution of plastics over the water systems, the mean plastic item density in Leiden and Wageningen was of similar magnitude (111 items/km vs. 133 items/km, respectively). In Wageningen, most plastic items were found on the northwest side of the city center. In Leiden, potential sources such as supermarkets and restaurants are more distributed over the city center. In addition, the canal system in Leiden runs through the center, rather than just around it. Plastic is distributed more easily across the water system.

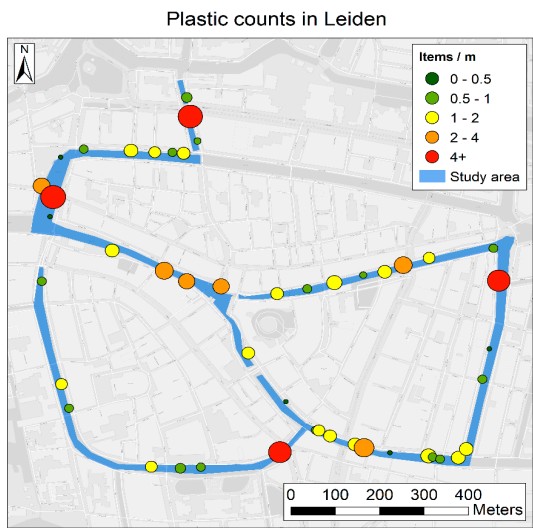

**Figure 4.** Plastic item density in the Leiden urban water system, 6 March 2020.

## Plastic counts in Wageningen

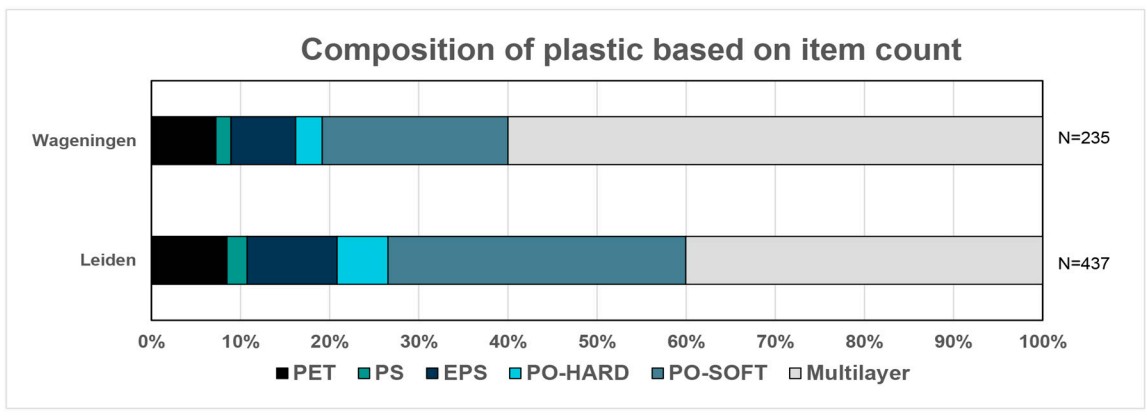

**Figure 5.** Plastic item density in the Wageningen urban water system, 5 March 2020.

In Leiden and Wageningen, most items were identified as Multilayer (Figure 6). However, the share of Multilayer to the total varied from 40% in Leiden to almost 60% in Wageningen. PO-soft was the second largest category in both cities. These categories combined accounted for 75% and 80% of the total in Leiden and Wageningen, respectively. Multilayer items are mainly associated with food wrappings and sachets, and PO-soft with plastic bags. The differences may be explained by the different sources in both cities. The results suggest that most items in Wageningen are related to unsoundly disposed food wrappings and sachets, which is supported by the hotspots close to the parking and recreational areas. In Leiden, a larger share of PO-soft can be explained by the market, which is situated next to the canal, as plastic bags are frequently used here to carry sold goods.

**Figure 6.** Plastic item categories for Leiden and Wageningen on 6 and 5 March 2020, respectively. Note that no observed items were classified as "other".

In both cities, it was found that plastic items tend to accumulate around locations where the water flow is obstructed. For example, increased densities were found close to recreational boats, houseboats, quay walls, dead ends of the canals, bridges, and between water lilies. Plastics that are either deposited or transported to such locations are likely to be retained for longer periods. The

anthropogenic influence on plastic mobilization, deposition, and transport was also clear in both cities. Hotspots and other locations with increased densities were close to potential sources, such as markets, locations related to student activities, terrace boats from restaurants, and recreational areas.

In this study, we expressed plastic density in items/m and items/km canal length, as this is a common unit to express riverbank plastic data (Vriend et al., 2020). Future studies may consider also normalizing over the canal or river width, allowing the expression of plastic density in items per surface area of the water system. This would allow for better comparison with other studies and locations, and give additional insight into hotspots and potential cleanup locations.

For long-term monitoring, we further recommend to explicitly report zero counts in segments, and to consistently measure the same segments. Our assessments primarily focused on identifying plastic hotspots. Long-term monitoring will also reveal under what conditions specific segments contain no plastics. Such insights will contribute to a better understanding of the transport mechanisms of plastics in urban water systems.

## 3.2. Comparison with Other Urban Water Systems

Direct comparison of the spatial distribution of plastic litter in the urban water system is difficult, because of the lack of observations. However, increased plastic litter amounts close to the sources or at hydraulic infrastructures have been reported in previous work (e.g., Gasperi et al. [11], Honingh et al. [6], Weideman et al. [9]). The spatial distribution of plastic in Leiden and Wageningen suggests increased densities around sources (market, recreational area, parking lot), and locations where litter can be entrapped. The plastic polymer categories in Leiden and Wageningen are consistent with observations from other urban water systems. However, it should be recognized that all observations are only a snapshot at a certain point in time. Globally, PO-soft and Multilayer are the most observed polymer categories (e.g., van Calcar and van Emmerik [10], Lahens et al. [13]). This is also the case in Leiden and Wageningen, although the portion is higher than the global average (75–80% vs. 50%). PET is generally a small category, which was also found in this study (4% global vs. 8–10% in Leiden/Wageningen). The main goal of this paper is to present a method for plastic hotspot mapping in urban areas. To date, the application of this method has been limited. This method has been applied to only two cities, which were both measured for only a single day. Longer-term observations will give additional insights into the sources, sinks, and transport dynamics of plastic litter in urban water systems. Plastic litter accumulates around, for example, recreational boats, houseboats, bridges, water plants (e.g., yellow and white water lilies), floating gardens, and bird nests [25]. In some cases, it is not possible to accurately count all plastic items due to obstructed view or other practical difficulties. At the same time, such locations may also be used as proxies for the overall abundance of plastics in the urban water system. If the overall plastic density in the system is low, the accumulation in these areas will most likely also be low. Future monitoring strategies may focus on collecting data mainly at locations with natural accumulation, rather than collecting data in the entire water system. In similar spirit, such locations may also be the most effective locations for plastic collection.

## 3.3. Outlook

The proposed method for mapping plastic in urban water systems provides a blueprint for long-term monitoring and rapid assessments in other cities. Due to its simple nature, data collection may also be done through citizen science efforts. Litter on Dutch and German riverbanks is already being monitored by trained citizens (e.g., Kiessling et al. [18], Vriend et al. [19]). New mobile applications, such as CrowdWater [20,26], have been developed to facilitate citizen science plastic data collection in urban and natural water systems. These apps can be used for longer-term plastic hotspot mapping in urban water systems. Recent work demonstrated that plastic data collected by untrained citizens using the CrowdWater app in the Rhine (the Netherlands) and the Klang (Malaysia) rivers yielded similar estimates of plastic abundance, spatiotemporal variation, and polymer categories compared to the standard methods. Additionally, Falk-Andersson et al. [27] found data collected

by citizen science protocols on Norwegian beaches are comparable with data collected by OSPAR monitoring. Yet, counting plastic items by merely looking at what is visible from riverbanks is a limited methodology in attaining accurate plastic hotspot locations. Therefore, citizen science data collection may also be combined with existing and planned cleanup efforts in urban areas. In Leiden, several initiatives such as CanalCups and Plastic Spotter focus on removing plastics from the urban water system. Simultaneous data collection using physical sampling and visual observations may give additional insights into plastic hotspots in urban water systems, which is, in turn, beneficial for determining the sources of plastics.

This simple method facilitates longer-term data collection, which is necessary to further study the variation in plastic hotspots. The location and magnitude of hotspots are assumed to be influenced by variation in input, transport dynamics, and potential removal efforts. Plastic input into the urban water system is related to specific events (e.g., market days, festivals), weather conditions, and policy. Long-term data can be used to explore the effect of bans on specific items. Transport dynamics are mainly determined by the flow direction and velocity in the urban water system. As water systems in cities like Leiden and Wageningen are managed, understanding the relation between transport and water flow is important to minimize negative impact and optimize collection efforts.

## 4. Conclusions

This paper presents a simple method for macroplastic hotspot mapping in urban areas. Through visual observations, both plastic quantity and composition can be rapidly determined. Within several hours, water systems in urban city centers can be monitored. Initial applications in the Dutch cities of Leiden and Wageningen revealed similar mean plastic densities in the urban water system (111–133 items/km). Higher plastic densities were associated with recreational areas, car parking, market locations, and infrastructure. Plastic was more equally distributed in Leiden, compared to Wageningen. This is mainly related to the extent of the water system, and the spatial distribution of human activities. The plastic composition was similar in Leiden and Wageningen, with the majority of the plastics (75–80%) identified as soft items, food packaging, foils, and bags (PO-soft and Multilayer). This is comparable to other urban areas globally and suggests that local consumers are the main source of plastics in urban water systems. With this paper, we aim to show that relatively simple methods can yield valuable data on the abundance, distribution, and composition of macroplastic pollution in urban water systems. The method presented in this paper can easily be applied to other cities for rapid or long-term data collection efforts in a harmonized way. Future work will give additional insights in the applicability of this method to other urban areas, such as cities with high population density and suboptimal waste infrastructure, and cities with different types of urban water systems. Observations of plastic pollution in urban water systems are crucial to better understand and quantify an important source of plastic litter in rivers and oceans.

**Supplementary Materials:** The following are available online at http://www.mdpi.com/2076-3263/10/9/342/s1, Data S1: Plastic hotspot observations Leiden and Wageningen.

**Author Contributions:** Conceptualization: P.T., H.Z., T.v.E.; Methodology: P.T., H.Z., T.v.E.; Formal Analysis: P.T., H.Z.; Investigation: P.T., H.Z.; Data Curation: P.T., H.Z.; Writing—original draft preparation: P.T., H.Z., T.v.E.; Writing—review and editing: all authors; Visualization: P.T., H.Z., D.S. All authors have read and agreed to the published version of the manuscript.

**Funding:** This research received no external funding.

**Acknowledgments:** We thank Leiden University and Leiden Municipality for their support of the Plastic Spotter project.

**Conflicts of Interest:** The authors declare no conflict of interest.

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
