# Peer review of "Plastic Hotspot Mapping in Urban Water Systems"

_geosciences, doi:10.3390/geosciences10090342_

Round 1

Reviewer 1 Report

This paper present an easy method to map and quantify plastics in surface water.  The topic and research is timely and important on a global scale.  I have reservations on the lack of controls and quality assurance in interpreting the results.  I find the conclusions fairly general without any comparison to controls.  For example, the authors did not identify how the differences between the two cities can be utilized in their analytical methods other than comparing plastics waste management practices.

Author Response

We thank the reviewers for their valuable suggestions that helped to improve the paper. We closely followed the comments in preparing the revised manuscript. We addressed all comments one by one, with our reply in bold.

---

Reviewer 1

  1. This paper present an easy method to map and quantify plastics in surface water.  The topic and research is timely and important on a global scale. 

Thank you for emphasizing the timeliness and relevance of our study.

  1. I have reservations on the lack of controls and quality assurance in interpreting the results.  I find the conclusions fairly general without any comparison to controls.  For example, the authors did not identify how the differences between the two cities can be utilized in their analytical methods other than comparing plastics waste management practices.

One of the main challenges with macroplastic pollution observations is the lack of control measurements or baseline values. In our paper, we present the first observations of plastic pollution in the Leiden and Wageningen urban water system. Understanding plastic dynamics in these two cities is the start of what we hope to be a larger citizen-science guided step towards understanding plastic dynamics in urban water systems in general. Comparing plastic waste management practices is the one of the uses of the analytical method used in our study. When increased data is available on potential hotspot locations and their geography, it can also be used to improve future urban design and planning. Additionally, by understanding where plastics originate from, one can solve the leakage at the origin rather than fight the symptoms. For instance, when the occurrence of plastic cups and/or polystyrene trays from specific restaurant chains is suddenly increasing in an urban water system, it can be an initiative for those restaurant chains to invest in better on-site waste collection infrastructure. Moreover, when the locations for potential litter hotspots are known with higher accuracy, targeted clean-ups can be more efficient than conventional waste management practices.

Reviewer 2 Report

A brief summary:

The manuscript 'Plastic hotspot mapping in urban water systems' by Tasseron et al. details a method for counting floating macroplastics in urban river systems and detailing potential hotspots. The authors describe the method and then provide results from a test case of their method in two urban water systems in the Netherlands. The paper's main contribution is to detail an easy to follow method that anyone can utilize for measuring floating plastic waste in urban river systems.

Broad comments:

Strengths:

The paper clearly describes their method for floating macroplastic assessment, including very easy to understand figures and graphs. In addition, the author's test the method and clearly detail the results. The method they describe would be easy for any individual to implement, even those with little to no equipment and/or scientific training.

Scientific weaknesses:

The authors could do a better job pointing out the weaknesses in the method. In particular, their system for characterizing the specific plastic polymers is very crude, and is likely to lead to many false calls. This does not invalidate the method, as it out of the scope of any citizen assessment method to use more precise measurements, such as Fourier-transform infrared spectroscopy (FTIR). However, the authors need to make it more clear that they are aware of these limitations. Furthermore, they give no allowance for plastic that does not fit their five criteria. It would make more sense to include an 'unknown plastic category'.

In addition, this method only captures floating plastic visible from a river bank. Once again, that does not invalidate the method, but a section where the authors discuss the obvious limitations of the method, and ways to maybe improve on the method in the future, would benefit the manuscript.

Editorial weaknesses:

This reviewer found grammatical and editorial typos, as well as some unusual usage of punctuation. In particular, the authors use of parentheses before a word to suggest an alternative meaning was distracting, and in general, unnecessary. While in theory parentheses can be used in this way, it is such an uncommon usage that the reviewer had to look up whether this was grammatically correct. Bear in mind, the reviewer is a native English speaker. For example, the authors' referred to '(house)boats' multiple times in the manuscript. Why not just say house-boats? Whenever possible the authors should keep the language as clear and concise as possible.

In addition, certain sentences lacked crucial commas and there were several instances of repeated words and too many spaces between words. Grammatical errors like these can seem unimportant, but they distract the reader and give the manuscript the feeling of being less-than-professional. Whenever possible the reviewer strongly recommends having multiple people proof-read the final manuscript before submission.

Finally, an overall suggestion. During the writing and reviewing process, the authors should ask themselves whether all the information provided is necessary for the manuscript. Does saying 'house-boat' versus 'boat' provide information necessary for understanding this study? Are two types of water lilies as examples of water plants necessary for a scientific report on a new method of plastic assessment? Like the strange use of punctuation, extraneous information like this distracts the reader from the important parts of the scientific study and should be avoided.

Specific comments:

Abstract: Clearly puts forth the purpose and importance of the research.

31: Why is 'aquatic' in parentheses? This is a confusing use of the parentheses. Take it out.

36: Make it clear that when you say 'emission', you are referring to plastic traveling from the rivers to another body of water.

39: '....the plastic pollution challenge...', the authors should avoid colloquial phrasing.

39 - 42: The second and third sentence in this paragraph say the exact same thing.

47: What does 12.5% refer to? 12.5% of all birds and fish ingest plastic? Or 12.5% of all plastic is ingested by birds and fish? Please make this more clear.

50: Again with this strange use of parentheses. It's distracting. Either it is macro-plastics, or all plastics, be precise.

62: I am starting to realize the authors really like using parentheses. There are so many in this paragraph already and I am not even done with it. There is nothing wrong with parentheses in general, but they should be used rarely. The authors need to find a way of conveying information that is more direct. Since the information within the parentheses could be taken out of the sentence without losing meaning, the authors should ask themselves why they are including so much unnecessary information in their writing.

In English, it is more common to put the country after the city with a comma, such as Jakarta, Indonesia.

Instead of '(plastic) litter', which is very uncommon in English, try 'types of litter items, including plastic, abundant....'

63: 'of' should be 'but'

73: Either the 'for example' is missing commas on either side of it, or it is a typo.

79: Again, 'for example' needs commas on either side of it. It should look like this: 'Data collection can, for example, be done...' But also, the 'for example' is completely unnecessary here. The 'can be' that follows suggests that same thing.

90: 'UAVs' : define a term the first time you use it, instead of immediately using an abbreviations.

111: Figure 2: An efficient way of explaining the method that is easy to understand.

129: Either 'house-boats' or 'boats', or 'boats, including house-boats', but NOT '(house)boats'!

135: There is an extra space after 'when'.

136: There is an extra 'that' at the end of the line.

143: How accurate is categorizing plastic by 'structural characteristics'? What happens when a plastic piece isn't one of these obvious consumer goods? Do really all plastic fit into these categories? The authors should do a better job making it clear that this is a crude method for determining plastic composition and is likely to mischaracterize items on a relative basis.

149: There is an extra space after 'systems'.

163: Is 'dataseries' meant to be a single word? I don't think it is a word.

164: The use of 'Besides' at the beginning of the sentence is confusing. Take it out and just start the sentence with 'Different'.

182: 'Furthermore, this supports reducing plastic pollution at its source.' Is a confusing sentence. Do the authors mean to say, 'This type of information would help reduce plastic pollution at its source."?

199: Didn't the authors say earlier that multi-day sampling leads to plastic being counted twice?

210: 'Most likely' isn't appropriate phrasing for a scientific study. Perhaps state that this is a hypothesis of the authors, or an observation.

229: Remove the parentheses around 'house'. See above for suggestions on how to rephrase this.

257: '(house)boats' again, see above

258: Why are only two types of water lilies given as examples of water plants? Comes off as strange and distracting. How is this important information? Are two different colors of water lilies really the only two water plants plastic could accumulate around? Recommend removing this parentheses, and the information contained within entirely.

267: Another inappropriate use of parentheses. For a native English speaker, it jarring to see punctuation used so strangely. Just say 'rapid assessments'.

269: Again! Just say 'trained citizens'! Whenever possible, go with the most precise wording possible.

281: Again, 'for example' requires commas on either end of it if it is put in the middle of a sentence. '...the effect of, for example, bans...'. However, the authors should consider not putting 'for example' in the middle of sentences so many times within one manuscript. The inclusion of 'can be' earlier in the sentence makes the 'for example' superfluous. It reads better as 'Long-term data can be used to explore the effect of bans on specific items.'

Supplemental table: The 'additional comments' are in Dutch and English. I would recommend making sure the whole table is in English, just in case someone else opens it and tries to read the comments. 

Author Response

We thank the reviewers for their valuable suggestions that helped to improve the paper. We closely followed the comments in preparing the revised manuscript. We addressed all comments one by one, with our reply in bold.

---

Reviewer 2

  1. This article is about surveying litter in canals, and has the potential to be a useful contribution to the literature. However, in its current form there seems to be a disconnect between what is stated in the abstract, and what is in the main body of the manuscript.

Thank you for your constructive feedback.

  1. The abstract says that there will be a method described, however the description of the method if rushed and incomplete. To remedy this, the authors should describe further, and in more detail the consequences of each decision made along the seven steps described in figure 2. Especially within the case studies.

We improved the description of the method, and provided additional support for the specific choices made in the data collection in Leiden and Wageningen.

  1. What makes this method so useful tor citizen science? I think it is necessary to make a case that this method is both rapid and accurate. At the moment these is no evidence that this is the case.

A recent study showed that crowd-sourced macroplastic observations yielded similar estimates of plastic abundance, spatiotemporal variation and polymer categories compared to the standard methods. Our plastic hotspot mapping method uses the same measurement technique, and is therefore assumed to be suitable for citizen science data collection. We added:

“Recent work demonstrated that plastic data collected by untrained citizens using the CrowdWater app in the Rhine (the Netherlands) and the Klang (Malaysia) rivers yielded similar estimates of plastic abundance, spatiotemporal variation and polymer categories compared to the standard methods.”

Detailed comments below:

  1. It is not clear if the whole canal was surveyed, or if sites were selected. If the later, there is not mention of the method of site selection. This needs to be made clear.

Thank you for your comment on this. In Figure 4 and 5 (the maps of plastic item counts of the case studies) the light blue area indicates the surveyed area. All 5-meter canal segments were marked as a measurement point, wherein an area of 5 meters along the canal length and over the whole canal width items were counted. This implies, if there is no item count point on the maps, the waterway on the 5-meter in segment was clear of any floating plastics. We’ve added the following line to made clear that the whole canal was surveyed: “If no plastics were counted within a 5-meter canal segment, the area is clear and therefore not marked as a measurement point with a zero plastic count.”

  1. The first paragraph in the methods section is repetitive of the introduction, and is not really describing methods.

We now use this paragraph as a short summarizing intro, and explain that we describe the methods in the following sections.

  1. Line 134 – please clarify what if meant by weather conditions did not transport litter at the time of sampling, yet you had protocol for when litter was moving? And later in the manuscript mention accumulation areas due to wind.

We omitted this sentence, as we agree it was unclear. The protocol used for data collection assumed negligible plastic item transport during the measurement. However, in case this did occur plastic items were counted in the segment where they were observed. We mention wind later in the manuscript as possible explaining factor for plastic hotspot formation.

  1. Line 136 – “That that” should be “at that”

Corrected.

  1. Section 2.3.6 is not clear. How are hot spots defined and then found from these data?

The definition of a ‘hotspot’ is arbitrary and depends on the level of pollution in an urban waterway system. As mentioned in the first line of section 3.1, hotspots for our case studies are defined when the plastic count exceeds 4 items per meter. Since we use 5-meter canal segments as our measurement points, the counts in a 5-meter segment had to be at least 20 to classify as a ‘hotspot’. We have modified section 2.3.6 (now 2.2.6) to give the definition of a hotspot and how these can be extracted from the data.

  1. Items per meter may not an appropriate metric of reporting litter densities. Rivers are not the same width throughout therefore items per area should be reported. What is the width of the canals sampled? This would allowing the interpretation of the area surveyed. For people unfamiliar with canals vs rivers, this is vital information.

Thank you for this valid comment. The width of waterways in Wageningen was smaller than the waterways in Leiden. There is still no common, harmonized unit to report plastic abundance, transport, concentrations or density in specific parts of water systems. Riverbank plastic pollution is mainly reported in items/km riverbank, plastic transport in items/hour (for the total river width). For plastic floating in relatively static urban waterways, the unit items/m or items/km canal length was considered most appropriate. Normalizing over the canal width may result in values that are not directly meaningful for stakeholders, as they are mainly interested in what canal segments have highest plastic abundance, and not necessarily concentrations. For other purposes (e.g. optimizing cleanup) concentration maps (items/m2) may provide additional insights however. We made changes to section 2.2.2 explaining why the items/m worked for our case studies  and what can be done in case waterways get significantly wider.

  1. Section 2.3.7 is also vague, it reads as though this (along with section 2.3.6) are “tacked on” without serious though about what is useful and needed for another researcher to use this method.

Thank you for your comment. We have changed section 2.3.6 (now 2.2.6) to give a more clear definition of plastic hotspots and discarded some sentences with redundant information. We have partially re-written section 2.3.7 (now 2.2.7) to make it a more integrated part of the methods, rather than being tacked on.

  1. the whole text should be checked for typos: there is inconsistent referencing style and incorrect figure references.

Thank you for pointing this out. We corrected the figure references and applied spelling and grammar checks throughout the entirety of the text.

  1. I also suggest the supplementary materials be rearranged for clarity and appropriate metadata added to allow use of these data in the future.

Supplementary materials have been cleaned up for clarity with all data needed to reproduce the figures as portrayed in our study. All comments translated to English.

Reviewer 3 Report

This article is about surveying litter in canals, and has the potential to be a useful contribution to the literature. However, in its current form there seems to be a disconnect between what is stated in the abstract, and what is in the main body of the manuscript.

The abstract says that there will be a method described, however the description of the method if rushed and incomplete. To remedy this, the authors should describe further, and in more detail the consequences of each decision made along the seven steps described in figure 2. Especially within the case studies. What makes this method so useful tor citizen science? I think it is necessary to make a case that this method is both rapid and accurate. At the moment these is no evidence that this is the case.

Detailed comments below:

It is not clear if the whole canal was surveyed, or if sites were selected. If the later, there is not mention of the method of site selection. This needs to be made clear.

The first paragraph in the methods section is repetitive of the introduction, and is not really describing methods.

Line 134 – please clarify what if meant by weather conditions did not transport litter at the time of sampling, yet you had protocol for when litter was moving? And later in the manuscript mention accumulation areas due to wind.

Line 136 – “That that” should be “at that”

Section 2.3.6 is not clear. How are hot spots defined and then found from these data? Items per meter may not an appropriate metric of reporting litter densities. Rivers are not the same width throughout therefore items per area should be reported. What is the width of the canals sampled? This would allowing the interpretation of the area surveyed. For people unfamiliar with canals vs rivers, this is vital information.

Section 2.3.7 is also vague, it reads as though this (along with section 2.3.6) are “tacked on” without serious though about what is useful and needed for another researcher to use this method.

the whole text should be checked for typos: there is inconsistent referencing style and incorrect figure references.

I also suggest the supplementary materials be rearranged for clarity and appropriate metadata added to allow use of these data in the future.

Author Response

We thank the reviewers for their valuable suggestions that helped to improve the paper. We closely followed the comments in preparing the revised manuscript. We addressed all comments one by one, with our reply in bold.

---

Reviewer 3

A brief summary:

  1. The manuscript 'Plastic hotspot mapping in urban water systems' by Tasseron et al. details a method for counting floating macroplastics in urban river systems and detailing potential hotspots. The authors describe the method and then provide results from a test case of their method in two urban water systems in the Netherlands. The paper's main contribution is to detail an easy to follow method that anyone can utilize for measuring floating plastic waste in urban river systems.

Thank you for your positive and constructive feedback.

Broad comments:

Strengths:

  1. The paper clearly describes their method for floating macroplastic assessment, including very easy to understand figures and graphs. In addition, the author's test the method and clearly detail the results. The method they describe would be easy for any individual to implement, even those with little to no equipment and/or scientific training.

Thank you for pointing out these strengths.

Scientific weaknesses:

  1. The authors could do a better job pointing out the weaknesses in the method. In particular, their system for characterizing the specific plastic polymers is very crude, and is likely to lead to many false calls. This does not invalidate the method, as it out of the scope of any citizen assessment method to use more precise measurements, such as Fourier-transform infrared spectroscopy(FTIR). However, the authors need to make it more clear that they are aware of these limitations. Furthermore, they give no allowance for plastic that does not fit their five criteria. It would make more sense to include an 'unknown plastic category'.

Thank you for your comment. A trade-off exists between the easiness to obtain data (citizen science-based) and the quality and accuracy of the dataset. In their paper, Falk-Andersson et al. (2019) state: “Being dependent on volunteers, data collected through citizen science may be affected by the volunteers not being familiar with the categories in the protocol, they may only partly clean and register the beach, and some categories may be more likely than others to not be properly registered, or even registered at all.” , which justifies your concerns about the plastic categorisation accuracies. In the Results & Discussion section, we have made changes to emphasize the inaccuracies caused by the choices of volunteer citizens. Additionally, we added the ‘other’ category to make misclassifications less likely if citizens are uncertain what type of plastics they are looking at.

  1. In addition, this method only captures floating plastic visible from a river bank. Once again, that does not invalidate the method, but a section where the authors discuss the obvious limitations of the method, and ways to maybe improve on the method in the future, would benefit the manuscript.

We have made changes in section 2.2.2 and the Results & Discussion to further discuss the limitations and possible steps for improvements of the methods used in our paper.

Editorial weaknesses:

  1. This reviewer found grammatical and editorial typos, as well as some unusual usage of punctuation. In particular, the authors use of parentheses before a word to suggest an alternative meaning was distracting, and in general, unnecessary. While in theory parentheses can be used in this way, it is such an uncommon usage that the reviewer had to look up whether this was grammatically correct. Bear in mind, the reviewer is a native English speaker.

    Thank you for noticing the frequent use of parentheses in our writing style. We generally use parentheses as additional information to render a holistic image of the situation or give examples. We have taken your comment into account for the entirety of our study.

    For example, the authors' referred to '(house)boats' multiple times in the manuscript. Why not just say house-boats? Whenever possible the authors should keep the language as clear and concise as possible.

    The Dutch urban water systems and canals are characterised by multiple types of boats. Recreational boats stationed at quay walls and docks are supplemented by permanent, stationary boats in which people live. The different boat types influence plastic flow dynamics in the water systems differently.

  1. In addition, certain sentences lacked crucial commas and there were several instances of repeated words and too many spaces between words. Grammatical errors like these can seem unimportant, but they distract the reader and give the manuscript the feeling of being less-than-professional. Whenever possible the reviewer strongly recommends having multiple people proof-read the final manuscript before submission.

Thank you for your commenting on the structural and grammatical errors in our manuscript. We applied several grammar and spelling checks on the final manuscript and it has been proof-read by additional people.

  1. Finally, an overall suggestion. During the writing and reviewing process, the authors should ask themselves whether all the information provided is necessary for the manuscript. Does saying 'house-boat' versus 'boat' provide information necessary for understanding this study?

See our explanation under comment 6.

  1. Are two types of water lilies as examples of water plants necessary for a scientific report on a new method of plastic assessment? Like the strange use of punctuation, extraneous information like this distracts the reader from the important parts of the scientific study and should be avoided.

These types of water lilies are the two types of water plants typically found in the canals of Leiden. These exact types of plants entrap floating litter and are therefore important to mention. They are not only found in the canals of Leiden, but also in Amsterdam and other Dutch waterways. We think it is important to be specific about plant species in this case.

 Specific comments:

 Abstract: Clearly puts forth the purpose and importance of the research.

 Thank you.

  1. 31: Why is 'aquatic' in parentheses? This is a confusing use of the parentheses. Take it out.

 Corrected. The focus of our study is on aquatic environments, yet plastic pollution in other environments also negatively impacts livelihood and ecosystem health.

  1. 36: Make it clear that when you say 'emission', you are referring to plastic traveling from the rivers to another body of water.

 Thank you for this suggestion. Changed to ‘...emission to other water bodies...’

  1. 39: '....the plastic pollution challenge...', the authors should avoid colloquial phrasing.

Corrected.

  1. 39-42: The second and third sentence in this paragraph say the exact same thing.

Rephrased.

  1. 47: What does 12.5% refer to? 12.5% of all birds and fish ingest plastic? Or 12.5% of all plastic is ingested by birds and fish? Please make this more clear.

It refers to the percentage of the analyzed birds and fish than contained plastics. We have clarified this in the text.

  1. 50: Again with this strange use of parentheses. It's distracting. Either it is macro-plastics, or all plastics, be precise.

Parentheses removed, since the focus of our study is on macroplastics 

  1. 62: I am starting to realize the authors really like using parentheses. There are so many in this paragraph already and I am not even done with it. There is nothing wrong with parentheses in general, but they should be used rarely. The authors need to find a way of conveying information that is more direct. Since the information within the parentheses could be taken out of the sentence without losing meaning, the authors should ask themselves why they are including so much unnecessary information in their writing.

Thank you for noticing the frequent use of parentheses in our writing style. We generally use parentheses as additional information to render a holistic image of the situation or give examples. We have taken your comment into account for the entirety of our study.

  1. In English, it is more common to put the country after the city with a comma, such as Jakarta, Indonesia.

 Thank you for noticing this. We decided to leave the sentences with cities and countries unchanged to preserve the readability of the sentences, and prevent large amount of commas in one sentence.

  1. Instead of '(plastic) litter', which is very uncommon in English, try 'types of litter items, including plastic, abundant....'

 Thank you for pointing this out. Corrected following your suggestion.

  1. 63: 'of' should be 'but'

 Corrected.

  1. 73: Either the 'for example' is missing commas on either side of it, or it is a typo.

‘For example’ removed.

  1. 79: Again, 'for example' needs commas on either side of it. It should look like this: 'Data collection can, for example, be done...' But also, the 'for example' is completely unnecessary here. The 'can be' that follows suggests that same thing.

 ‘For example’ removed.

  1. 90: 'UAVs' : define a term the first time you use it, instead of immediately using an abbreviations.

 Term defined without immediately using abbreviation.

  1. 111: Figure 2: An efficient way of explaining the method that is easy to understand.

 Thank you for pointing out its efficiency.

  1. 129: Either 'house-boats' or 'boats', or 'boats, including house-boats', but NOT '(house)boats'!

Thank you for pointing out our superfluous use of parentheses again. Corrected to ‘recreational boats and house-boats'.

  1. 135: There is an extra space after 'when'.

 Corrected.

  1. 136: There is an extra 'that' at the end of the line.

 Corrected.

  1. 143: How accurate is categorizing plastic by 'structural characteristics'? What happens when a plastic piece isn't one of these obvious consumer goods? Do really all plastic fit into these categories? The authors should do a better job making it clear that this is a crude method for determining plastic composition and is likely to mischaracterize items on a relative basis.

Thank you for your comment on one of the weaknesses of our method. It is indeed hard to categorise plastic by structural characteristics which can lead to mischaracterisations. However, due to the simple nature and its potential application for citizen science-based data collection, it is necessary to use categories which can be recognized by looking at plastics from quays and streets next to canals. For higher classification accuracy, collection of plastic from surface water and subsequent spectral analyses in laboratories is required, which is not compatible with the citizen science concept. In our results and discussion section we reflect on multiple aspects of the limitations and (in)accuracy of classification.

  1. 149: There is an extra space after 'systems'.

 Corrected.

  1. 163: Is 'dataseries' meant to be a single word? I don't think it is a word.

 Thank you, I think we tried to incorrectly combine ‘timeseries’ with ‘data’ here. Corrected to ‘datasets’

  1. 164: The use of 'Besides' at the beginning of the sentence is confusing. Take it out and just start the sentence with 'Different'.

 Corrected following your suggestion.

  1. 182: 'Furthermore, this supports reducing plastic pollution at its source.' Is a confusing sentence. Do the authors mean to say, 'This type of information would help reduce plastic pollution at its source."?

 Thank you, we agree it is a confusing sentence. Corrected following your suggestion.

  1. 199: Didn't the authors say earlier that multi-day sampling leads to plastic being counted twice?

Rephrased.

  1. 210: 'Most likely' isn't appropriate phrasing for a scientific study. Perhaps state that this is a hypothesis of the authors, or an observation.

 We did indeed observe this, corrected to: “The hotspots are observed to be related to the parking....”

  1. 229: Remove the parentheses around 'house'. See above for suggestions on how to rephrase this.

 Corrected.

  1. 257: '(house)boats' again, see above

 Corrected.

  1. 258: Why are only two types of water lilies given as examples of water plants? Comes off as strange and distracting. How is this important information? Are two different colors of water lilies really the only two water plants plastic could accumulate around? Recommend removing this parentheses, and the information contained within entirely.

See explanation at comment 10. These types of water lilies are the two types of water plants typically found in the canals of Leiden. These exact types of plants entrap floating litter and are therefore important to mention. They are not only found in the canals of Leiden, but also in Amsterdam and other Dutch waterways. We think it is important to be specific about plant species in this case.

  1. 267: Another inappropriate use of parentheses. For a native English speaker, it jarring to see punctuation used so strangely. Just say 'rapid assessments'.

Parentheses removed.

  1. 269: Again! Just say 'trained citizens'! Whenever possible, go with the most precise wording possible.

Parentheses removed.  

  1. 281: Again, 'for example' requires commas on either end of it if it is put in the middle of a sentence. '...the effect of, for example, bans...'. However, the authors should consider not putting 'for example' in the middle of sentences so many times within one manuscript. The inclusion of 'can be' earlier in the sentence makes the 'for example' superfluous. It reads better as 'Long-term data can be used to explore the effect of bans on specific items.'

Thank you for your suggestion, the sentence is changed to “Long-term data can be used to explore the effect of bans on specific items.”

  1. Supplemental table: The 'additional comments' are in Dutch and English. I would recommend making sure the whole table is in English, just in case someone else opens it and tries to read the comments. 

Thank you for your comment. We ‘cleaned’ the table in general, so it only contains the data that was used in making the graphs and figures. All comments are now in English.

Round 2

Reviewer 1 Report

None

Author Response

N/A

Reviewer 3 Report

The authors have responded well to the original review, and the manuscript is improved. However, I still have two small comments:

I would argue that the justification of items per metre is not ideal, and a per area measure would give a more comparable (with other studies and sites) and accurate report of litter abundance. I agree that standards are not set in the literature, but we should be aiming for the best practice possible. I did agree that local government or NGOs could not use a per area unit.

I also believe that areas with zero count are important and valuable pieces of data and should not be neglected, or pooled with small counts as in figures 4 and 5.

Author Response

We thank the reviewer for their additional suggestions. We addressed all comments one by one, with our reply in bold.

---

The authors have responded well to the original review, and the manuscript is improved. However, I still have two small comments:

Thank you for your additional feedback.

I would argue that the justification of items per metre is not ideal, and a per area measure would give a more comparable (with other studies and sites) and accurate report of litter abundance. I agree that standards are not set in the literature, but we should be aiming for the best practice possible. I did agree that local government or NGOs could not use a per area unit.

We agree, and added the following to the Results and Discussion:

“In this study we expressed plastic density in items/m and items/km canal length, as this is a common unit to express riverbank plastic data (Vriend et al., 2020). Future studies may consider also normalizing over the canal or river width, allowing to express the plastic density in items per surface area of the water system. This would allow for better comparison with other studies and locations, and give additional insight in hotspots and potential cleanup locations.”

I also believe that areas with zero count are important and valuable pieces of data and should not be neglected, or pooled with small counts as in figures 4 and 5.

We agree, and implemented this comment as additional recommendation:

“For long-term monitoring we further recommend to explicitly report zero counts in segments, and to consistently measure the same segments. Our assessments primarily focused on identifying plastic hotspots. Long-term monitoring will also reveal under what conditions specific segments contain no plastics. Such insights will contribute to a better understanding of the transport mechanisms of plastics in urban water systems.”